# Improved Satellite Cell Proliferation Induced by L-Carnosine Benefits Muscle Growth of Pigs in Part through Activation of the Akt/mTOR/S6K Signaling Pathway

Yaojun Liu [1], Wenqiang Shen [1], Tao Liu [1], Rainer Mosenthin [2], Yinghui Bao [1], Peng Chen [1], Wenbo Hao [1], Lihong Zhao [1], Jianyun Zhang [1], Cheng Ji [1] and Qiugang Ma [1,*]

1   State Key Laboratory of Animal Nutrition, China Agricultural University, Beijing 100193, China;
    sy20203040641@cau.edu.cn (Y.L.); shenwenqiang1996@163.com (W.S.); liuyitao2004@126.com (T.L.);
    byhui@126.com (Y.B.); chenpengcau@163.com (P.C.); haowenbocau@163.com (W.H.);
    zhaolihongcau@cau.edu.cn (L.Z.); jyzhang@cau.edu.cn (J.Z.); jicheng@cau.edu.cn (C.J.)
2   Institute of Animal Science, University of Hohenheim, 70593 Stuttgart, Germany;
    rainer.mosenthin@uni-hohenheim.de
*   Correspondence: maqiugang@cau.edu.cn

**Abstract:** (1) Background: L-carnosine (*β*-alanyl-L-histidine), a natural dipeptide, exists at relatively high concentrations in skeletal muscles, and has been shown to protect cells from adverse conditions due to its antioxidant, anti-aging, anti-glycation, and buffering properties. Satellite cells (SCs), residing on the myofiber surface, are crucial for muscle post-growth and regeneration. However, the effects of L-carnosine on muscle development of pigs in vivo, on proliferation and growth of SCs in vitro, and the relationship between SCs and muscle development have not yet been investigated. (2) Methods: The objective of this study was to assess the effect of dietary L-carnosine on growth performance and *longissimus dorsi* muscle development of pigs in vivo, and to elaborate its molecular mechanisms in vitro using L-carnosine-treated SCs. (3) Results: It was shown that L-carnosine supplementation (0.2 and 2 mM) increased ($p < 0.05$) SC proliferation and cell percentage in the synthesis (S) phase and decreased cell percentage in the resting (G0)/first gap (G1)/phase. Moreover, average daily gain (ADG) of pigs fed diets containing 0.1% of L-carnosine was higher ($p < 0.05$) than that of pigs fed diets without L-carnosine, and the *longissimus dorsi* muscle weight of pigs assigned to the L-carnosine treatments was 7.95% higher compared to control pigs. Both in the *longissimus dorsi* muscle and cultured SCs of pigs, the Akt/mTOR/S6K signaling pathway was activated ($p < 0.05$), suggesting that L-carnosine improved muscle growth and SC proliferation of pigs. (4) Conclusions: Considering the important role of SCs in post-natal muscle growth, there is evidence that L-carnosine may improve muscle growth of pigs through promoting SC proliferation via activating the Akt/mTOR/S6K signaling pathway.

**Keywords:** L-carnosine; satellite cells; proliferation; mTOR signaling pathway; muscle growth

## 1. Introduction

Skeletal muscles are highly adaptable and flexible tissues [1], and progressive muscle growth is a critical determinant for meat production from livestock. Muscle growth includes mainly two processes, namely hyperplasia and hypertrophy [2]. Myofiber changes in piglets mainly comprise myofiber enlargement, extension (hyperplasia), and fiber-type transformation [3,4]. Therefore, post-partum muscle growth in pigs depends on hypertrophic growth of the individual myofibers. During this growth phase, characterized by an increase in myofiber volume, they absorb satellite cell (SC) nuclei to maintain a comparatively constant ratio of nucleus to cytoplasm. SCs are a population of mononucleated stem cells with myogenic potential, and play an important role in post-natal muscle growth and regeneration [5,6]. In mature muscles, SC nuclei represent 3–6% of all muscle nuclei

and are usually mitotically quiescent, but can be activated by some events such as growth, regeneration, and injury [7]. Once activated, SCs will undergo multiple rounds of proliferation. Most of these cells differentiate and fuse into the existing myofibers to produce new myonuclei [8].

The mechanistic target of rapamycin (mTOR) represents a serine/threonine protein kinase, which is a pivotal regulator of protein synthesis, cell proliferation, and cell growth [9–12]. Notably, muscle fiber growth is closely associated with protein turnover, thus being engaged in the regulation of muscle mass growth [13]. The mTOR includes mTORC1 and mTORC2. In skeletal muscle, mTORC1 signaling can be activated by many stimuli, such as amino acids, cellular energy status, and insulin-like growth factor 1 (IGF-1) [14]. The mTORC2 is responsive to growth factors and regulates cell survival, apoptosis, growth, and proliferation [15].

Feed additives, nutrients that when added to the feed can trigger the desired response of the animal's body on production parameters, have been widely used in animal nutrition for quite a long time. In recent years, nutritional strategies have emerged and it has been well demonstrated that they can increase the level of production and improve the health of animals and products obtained from them [16], as well as to enhance livestock productivity [17,18]. There is evidence that L-carnosine supplementation ameliorates skeletal muscle protein loss mainly by maintaining proteostasis redox homeostasis in skeletal muscles [19]. In addition, L-carnosine may be beneficial in maintaining higher metabolic activity of muscle fibers [20], both under in vivo [21,22] and in vitro conditions [23]. It has been shown that L-carnosine has a positive effect on regulating specific changes in skeletal muscle, such as a decrease in protein synthesis with age [24], or a drastic metabolic response to exercise [25]. However, the communication mechanisms between L-carnosine, the mTOR pathway, SCs, and muscle growth are still unknown. It is hypothesized that supplemental L-carnosine may increase skeletal SC proliferation through the mTOR pathway, thus promoting muscle development of pigs. In view of such considerations, the aim of the current study was to evaluate potential effects of L-carnosine on growth performance and on the expression of pivotal regulatory factors of the mTOR pathway in the *longissimus dorsi* muscle of pigs. In addition, its effect on SC proliferation, the cell cycle of SCs, and the mTOR signaling pathway in cultured cells was studied.

## 2. Materials and Methods

### 2.1. Handling and Feeding Pigs

All animals enrolled in this study were determined to have a normal baseline health status. A total of 60 15-week-old castrated male Landrace pigs of approximately 50 kg body weight (BW) were allotted randomly to 5 dietary treatments. Each treatment included 12 replicates with 1 pig each. The 5 treatments consisted of a control diet (Table 1), either supplemented with 0.04% *β*-alanine (alanine diet, ALA) or 0.06% histidine (histidine diet, HIS) or 0.04% *β*-alanine+0.06% histidine (alanine and histidine diet, ALA+HIS) or 0.1% L-carnosine (L-carnosine diet). The control diet, based on corn and soybean meal, was formulated to meet the nutrient requirements for pigs according to National Research Council (NRC, 2012) recommendations.

The feeding trial was conducted at the Fengze pig farm (Fujian, China). Pigs were allowed ad libitum to obtain experimental diets and water. All animals were kept in pens individually. The experiment lasted for 5 weeks, corresponding to the pigs' final BW of about 75 kg. At end of the trial, average daily feed intake (ADFI), average daily gain (ADG), and feed: gain ratio were calculated. Pigs were slaughtered using electrical stunning 16 h after the last feeding in a commercial abattoir. Thereafter, they were exsanguinated, and area of loin eye at the 10th rib was recorded. For Western blot assay, the right side of the *longissimus dorsi* muscle was collected, frozen in liquid nitrogen immediately, and then stored at −80 °C.

**Table 1.** Ingredients and nutrient composition of the basal diet (% as-fed basis).

| Ingredients | % | Nutrition Component | Content |
|---|---|---|---|
| Corn | 46.90 | ME (MJ/kg) | 13.24 |
| Corn gluten meal | 5.70 | CP (%) | 22.7 |
| Extruded soybean | 20.00 | Ca (%) | 0.98 |
| Soybean meal | 20.00 | Total phosphorus (%) | 0.64 |
| Limestone | 1.20 | Available phosphorus (%) | 0.44 |
| Dicalcium phosphate | 1.70 | Met (%) | 0.50 |
| Salt | 0.30 | Met+Cys (%) | 0.90 |
| Corn oil | 2.70 | Lys (%) | 1.12 |
| Premix | 1.00 | ME (MJ/kg) | 13.24 |
| Bentonite | 0.50 | | |
| Total | 100.00 | | |

Note: The vitamin and mineral pre-mix supplied the following per kilogram of diet: vitamin A, 12040 IU; vitamin D3, 2112 IU; vitamin E, 29.7 IU; vitamin K3, 2.8 mg; vitamin Bl, 1.2 mg; vitamin B2, 7.1 mg; vitamin B6, 1.3 mg; vitamin Bl2, 0.03 mg; nicotinic acid, 42.9 mg; pantothenic acid, 21.6 mg; folic acid, 0.44 mg; biotin, 0.12 mg; choline chloride, 320 mg; Fe, 80 mg; Cu, 40 mg; Zn, 140 mg; Mn, 52 mg; Se, 0.33 mg; I, 0.75 mg.

### 2.2. Primary pig Skeletal Muscle SCs Culture

Skeletal muscle SCs were obtained from three 1-day-old Landrace piglets. Skeletal muscle SCs of pigs were separated using a slightly modified method described by Wang et al. [26]. Briefly, the *longissimus dorsi* muscle was excised, followed by careful removal of fat and connective tissue. Then, the scissors-sheared muscle was treated with 0.1% collagenase type II (Absin, Shanghai, China) for 2 h at 37 °C in a water bath. Subsequently, SCs of the pigs were separated from muscle fiber fragments and tissue debris by centrifugation and filtration, and further purified using the differential adhesion method [27]. The cells were cultured in DMEM/F-12 (Procell, Wuhan, China) supplemented with 1% penicillin/streptomycin, 2% chicken embryo extract, and 10% fetal bovine serum (FBS; Procell, Wuhan, China) (growth medium, GM). They were kept in a standard cell-culture incubator (MCO-18A, Panasonic, Japan) at 37 °C, and photographed for observation using a light microscope (Leica DM IL LED, Leica Microsystems GmbH, Wetzlar, Germany). The cell culture medium was renewed every second day. Immunocytochemistry analysis revealed that more than 95% of isolated SCs were positive for the rabbit anti-human paired box protein 7 (AF-7584, Affinity, Ann Arbor, MI, USA). Viability of the isolated SCs was also greater than 97%, as measured by an Invitrogen Countess 3 automated cell counter (Thermo Fisher Scientific, Rockford, IL, USA). Each cell culture experiment was based on three independent experiments which, in turn, were replicated using three different sources of cells isolated from three different piglets.

### 2.3. Proliferation Activity Analyses

2.3.1. MTT Assay

A pre-test trial was performed to determine the optimal concentration of L-carnosine to promote SC proliferation. The SCs were seeded in 96-well plates at a density of $1 \times 10^4$ cells/mL in GM without 2% chicken embryo extract. After seeding for 24 h, the cultures were treated with fresh GM containing a range of concentrations of L-carnosine (0, 0.2, 2, 10, 20, and 40 mM). The effects of L-carnosine on cell proliferation were determined at 24, 48, 72, 96, and 120 h after seeding. Briefly, 20 μL of 3-(4,5-dimethylthiazol-2-yl)-2,5-diphenyltetrazolium bromide (MTT) solution (5 mg MTT/mL PBS, Aladdin, Shanghai, China) was added to each well and incubated at 37 °C for 4 h. The mitochondrial enzymes in living cells have the capacity to transform MTT into insoluble formazan. The plates were centrifuged at $4000 \times g$ at 4 °C for 15 min. Thereafter, the supernatant from each well was carefully removed. To each well, 150 μL of dimethylsulfoxide (DMSO) working solution (180 mL DMSO with 20 mL 1 mol/L HCl) were added, and then the optical density (OD) value was measured by means of an ELISA reader (Thermo Scientific Multiskan SkyHigh, Rockford, IL, USA) at a wavelength of 490 nm.

### 2.3.2. Crystal Violet Assay

The SCs were cultured in a 96-well plate at a density of $1 \times 10^4$ cells/mL using a growth medium (GM, with 10% FBS). The cultures were treated with fresh GM supplemented with different concentrations of L-carnosine (0, 0.2, 2, and 20 mM) after seeding for 24 h. Proliferation of SCs was determined at 24, 48, 72, 96, and 120 h after seeding. Briefly, the plate was rinsed twice with PBS to remove residual GM and stained with 0.1% crystal violet solution for 2 h. After rinsing and drying the unstained crystal violet, 200 μL of 75% ethanol solution was added to each well, and then the OD value was measured by means of an ELISA reader at a wavelength of 570 nm.

### 2.3.3. Flow Cytometry

The SCs were cultured as previously described [28], with minor modifications, using a 6-well plate at a density of $7.5 \times 10^5$ cells/mL in GM. The cultured SCs were collected at 48, 72, and 96 h for subsequent analysis. The cells were fixed with 70% ethanol solution at 4 °C to determine the cell size. Subsequently, cells were rinsed twice with PBS, and run on a BD FAC Scan (Becton Dickinson, Franklin Lake, NJ, USA). The cell size was determined by measuring the forward scatter area (FSC-A).

For the cell-cycle analysis, the cultured cells were fixed in 2 mL ethanol at a temperature of 4 °C and stored at −20 °C. Thereafter, the cells were centrifuged at $1000\times g$ at 4 °C for 10 min, and then resuspended with 1 mL PBS. Finally, cells were treated with 200 μL propidium iodide (1 mg/mL) at room temperature for 15–20 min, and then subjected to flow cytometry (FCM) using a BD FAC Scan.

### 2.4. Western Blot

The SCs were washed twice with PBS at 4 °C and incubated on ice in a lysis buffer containing 1 mmol/L phenylmethanesulfonyl fluoride (Beyotime, Shanghai, China) and phosphatase inhibitors (Beyotime, Shanghai, China). Subsequently, the samples were centrifuged at $12,000\times g$ at 4 °C for 15 min. Total protein in the *longissimus dorsi* muscle was extracted using a T-PER Tissue Protein Extraction Kit (Absin, Shanghai, China). Protein concentration was measured using a BCA Protein Assay Kit (Absin, Shanghai, China). Protein samples were inactivated in boiled water for 10 min, and then 8 uL of sample was added into 10% sodium dodecyl sulfate (SDS) gel following the manufacturer's instructions (SDS-PAGE gel kit; Absin, Shanghai, China). The total proteins were first separated by electrophoresis at 80 V for 20 min and then the voltage was increased to 110 V for 75 min in Tris-glycine running buffer (0.025 mol/L Tris base, 0.192 mol/L glycine, and 0.1% SDS, pH 8.3). After completion of the electrophoresis, the protein samples in SDS-PAGE were transferred to polyvinylidene difluoride (PVDF) membranes (Absin, Shanghai, China) using semidry methodology. Membranes were probed using specific antibodies: anti-Akt (CST-9272), anti-phospho-Akt (ser473, CST-2211), anti-mTOR (CST-2972), anti-phospho-mTOR (Ser2448, CST-5536), anti-IGF1 (CST-73034), anti-70S6K (S6K, CST-2217), anti-phospho-70S6K (Thr389; CST-9205), anti-4EBP1 (CST-9452), anti-phospho-4EBP1 (Thr70; CST-9455), and anti-eIF4E(CST-2067). These primary antibodies were purchased from Cell Signaling Technology (Beverly, MA, USA), and the dilutions that were used for each antibody were produced according to manufacturer's instructions. Protein bands were visualized using ChemiDoc XRS+ (Bio-Red, Hercules, CA, USA) after hybridization with a horseradish peroxidase (HRP)-conjugated secondary antibody (diluted at 1:1000, Novus Biologicals, Littleton, CO, USA). Protein expression levels were normalized in relation to the level of the reference protein glyceraldehyde-3-phosphate dehydrogenase (GAPDH) or phosphor-protein. Quantification of relative protein expression was performed with ImageJ software (NIH, Bethesda, MD, USA). Representative Western blot bands were selected and statistical results were calculated from all the Western blot bands.

### 2.5. Statistical Analyses

For all data, conformity to normal distribution was determined using the Shapiro–Wilk test; all data passed the normal distribution test. Data between multiple groups were analyzed by one-way ANOVA followed by Duncan's multiple range test using SAS software (Version 8e, Cary, NC, USA). Welch's *t* test (normal distribution, unequal variance) or Student's *t* test (normal distribution, equal variance) was used for comparing differences between two groups. Data were considered significantly different at $p < 0.05$.

## 3. Results

### 3.1. Effect of L-Carnosine on Growth Performance of Finishing Pigs

Pigs fed the diet supplemented with 0.1% L-carnosine had higher ($p < 0.05$) ADFI than that of the HIS treatment (Table 2). In addition, the ADG of pigs in L-carnosine and ALA+HIS treatments was higher ($p < 0.05$) in comparison to the control and ALA and HIS treatments. There was no significant difference ($p > 0.05$) in ADG of pigs among the control, ALA, and HIS treatments. Although no significant difference in feed: gain ratio and loin eye area was observed between all treatments ($p > 0.0.5$), the loin eye area of pigs in L-carnosine treatments was numerically higher, amounting to 7.95% in comparison to pigs of the control treatment ($p = 0.321$).

**Table 2.** Effect of carnosine on growth performance of finishing pigs.

|  | Control | ALA | HIS | ALA+HIS | L-Carnosine | SEM | *p*-Value |
|---|---|---|---|---|---|---|---|
| ADFI (kg) | 2.86 [ab] | 2.72 [ab] | 2.49 [b] | 2.57 [ab] | 2.96 [a] | 0.14 | 0.036 |
| ADG (kg) | 0.773 [b] | 0.780 [b] | 0.769 [b] | 0.862 [a] | 0.875 [a] | 0.02 | 0.015 |
| Feed: gain ratio | 3.71 | 3.48 | 3.23 | 2.99 | 3.36 | 0.32 | 0.078 |
| Loin eye area (cm$^2$) | 38.74 | 39.35 | 38.65 | 40.67 | 41.82 | 2.34 | 0.321 |

Note: Mean values represent the mean of replicates ($n = 12$). ADFI, average daily feed intake; ADG, average daily gain; Control: diet without L-carnosine; ALA, diet supplemented with 0.04% $\beta$-alanine; HIS, diet supplemented with 0.06% L-histidine; ALA+HIS, diet supplemented with 0.04% $\beta$-alanine and 0.06% L-histidine; L-carnosine, diet supplemented with 0.1% L-carnosine. SEM, standard error of the mean. Means with different superscript letters indicate significant differences at $p < 0.05$.

### 3.2. Effect of L-Carnosine on the mTOR Signaling Pathway in the Longissimus Dorsi Muscle of Finishing Pigs

Pigs fed the diet supplemented with 0.1% L-carnosine had higher ($p < 0.05$) phosphorylation ratios of Akt (Ser473) (Figure 1A), mTOR (ser2448) (Figure 1B), and S6K (Thr389) (Figure 1C) than those fed the control diet (Figure 1). Supplementation of L-carnosine did not affect ($p > 0.05$) the phosphorylation ratio of 4EBP1 (Thr70) (Figure 1D).

### 3.3. Effect of L-Carnosine on Satellite Cell Proliferation

To investigate the effect of L-carnosine on piglets' SC proliferation, MTT assays and crystal violet assays were conducted (Figure 2A,B). According to the results of the pre-test trial, low concentrations of L-carnosine (0.2 and 2 mM) increased ($p < 0.05$) SC proliferation at 48, 72, 96, and 120 h after seeding in comparison to the control (Figure 2A,B). However, the SC proliferation was gradually inhibited by higher concentrations of L-carnosine, as supplementation of 20 or 40 mM of L-carnosine decreased ($p < 0.05$) SC proliferation at 72, 96, and 120 h (Figure 2A,B). At 120 h after seeding, high concentrations of L-carnosine (20 and 40 mM) resulted in a large number SCs floating and dying, which is in contrast to enhanced SC proliferation at low concentrations of L-carnosine (0.2 and 2 mM, Figure 2C).

Based on these results, concentrations of 0.2, 2, and 20 mM L-carnosine were chosen to assess the effect of L-carnosine or its precursors $\beta$-alanine and L-histidine on cell proliferation. Both MTT and the crystal violet assay revealed that similar to the results in the pre-test trial (Figure 2B), supplementation of L-carnosine at dosages of 0.2 and 2 mM improved ($p < 0.05$) SC proliferation at 72, 96, and 120 h after seeding as compared to the control (Figure 3A,B). Proliferation of SCs exposed to 20 mM was inhibited ($p < 0.05$) compared

to the L-carnosine at dosages of 0.2 and 2 mM (Figure 3A,B). It is known that L-carnosine can be metabolized into ALA and HIS. As biologically active molecules, *β*-alanine and L-histidine may influence SC proliferation. However, at low concentrations (0.2 and 2 mM), *β*-alanine and L-histidine alone or in combination did not affect ($p > 0.05$) SC proliferation based on results of MTT and crystal violet assay compared to the control (Figure 3C,D). On the contrary, when supplemented at a concentration of 20 mM, the combination of *β*-alanine and L-histidine inhibited ($p < 0.05$) SC proliferation at 72, 96, and 120 h after seeding as compared to the control (Figure 3A,B). According to the results of the MTT assay, the addition of a low concentration of *β*-alanine (0.2 mM) enhanced ($p < 0.05$) SC proliferation at 120 h after seeding as compared to the control; however, there was no effect ($p > 0.05$) at 24, 48, 72, and 96 h after seeding (Figure 3C). The proliferation of SCs was inhibited ($p < 0.05$) at a higher concentration of supplemental L-histidine (20 mM) at 48, 72, 96, and 120 h after seeding, but not at a low concentration of 2 mM (Figure 3C,D).

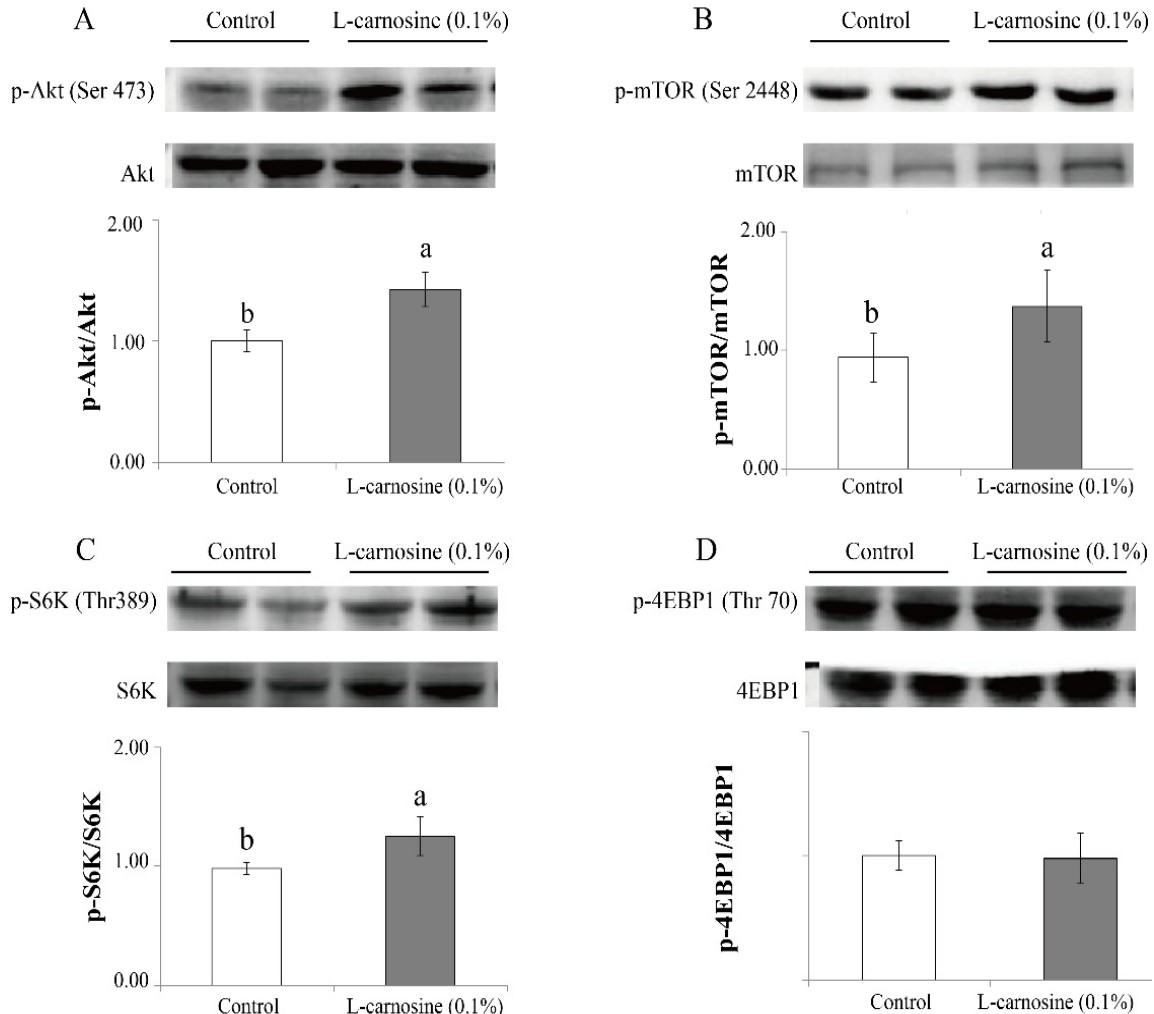

**Figure 1.** Effect of carnosine on expression of the mTOR pathway in the *longissimus dorsi* muscle of finishing pigs. Values of Akt (**A**), mTOR (**B**), S6K (**C**), and 4EBP1 (**D**) are presented as phosphorylation concentrations in relation to the total concentration. Control: diet without L-carnosine; L-carnosine (0.1%): diet supplemented with 0.1% L-carnosine. p-, phosphorylation; Ser, serine; Thr, threonine; Akt, protein kinase B; mTOR, mechanistic target of rapamycin; S6K, ribosomal protein S6 kinase; 4EBP1, 4E binding protein 1. Data are expressed as mean values ± SD. Means with different superscript letters (a, b) indicate significant differences at $p < 0.05$.

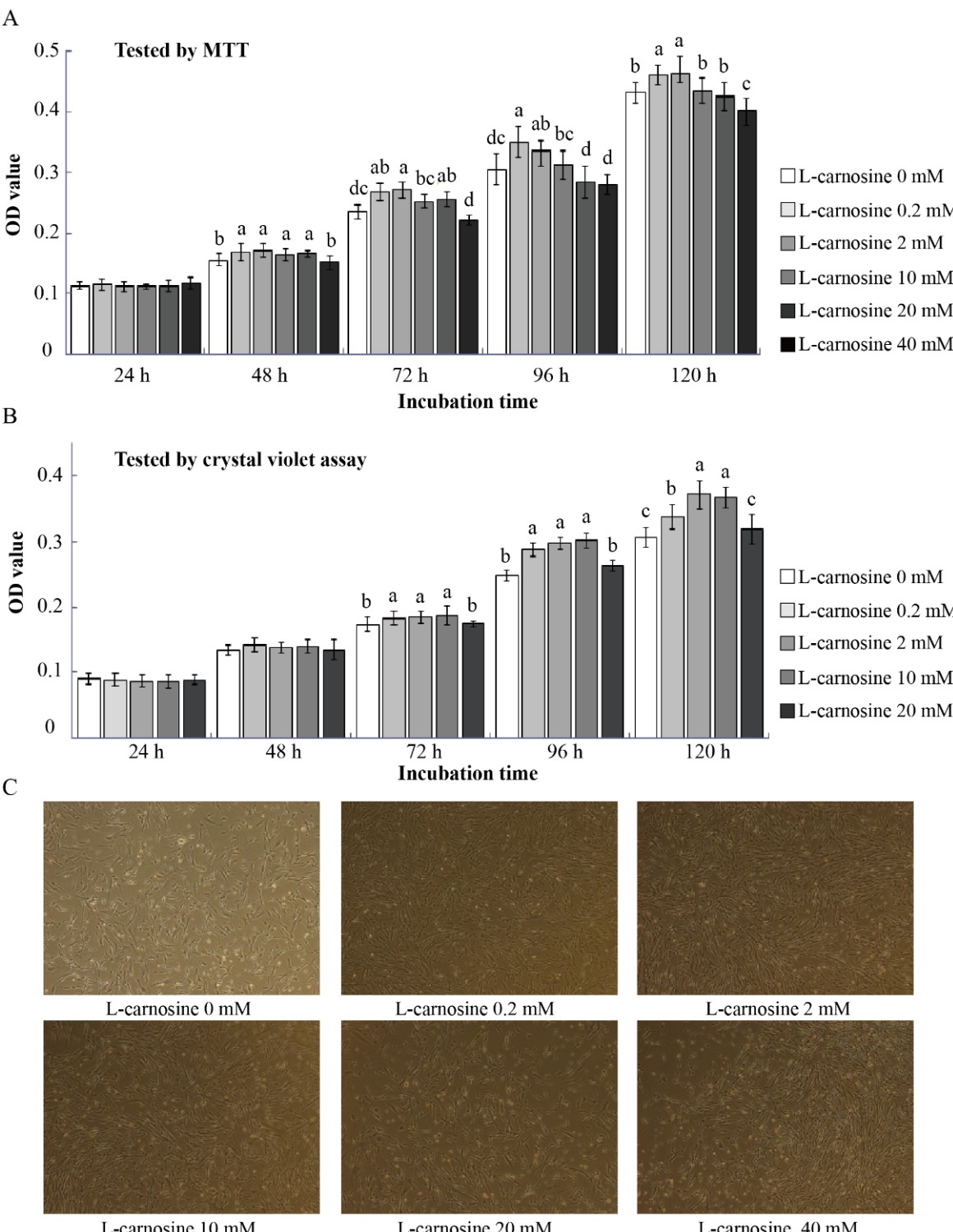

**Figure 2.** Effect of carnosine on the proliferation of SCs in vitro (*n* = 12). The effect of different concentrations of carnosine on cell proliferation (**A**,**B**); representative images of SCs treated with L-carnosine at 120 h (**C**, scale bar = 100 μm). MTT, 3-(4,5-dimethylthiazol-2-yl)-2,5-diphenyltetrazolium bromide; L-carnosine 0 mM, growth medium without L-carnosine; L-carnosine (0, 0.2, 2, 10, 20 and 40 mM), growth medium supplemented different concentrations of L-carnosine (0, 0.2, 2, 10, 20 and 40 mM). Data are expressed as mean values ± SD. Means with different superscript letters (a–d) indicate significant differences at *p* < 0.05.

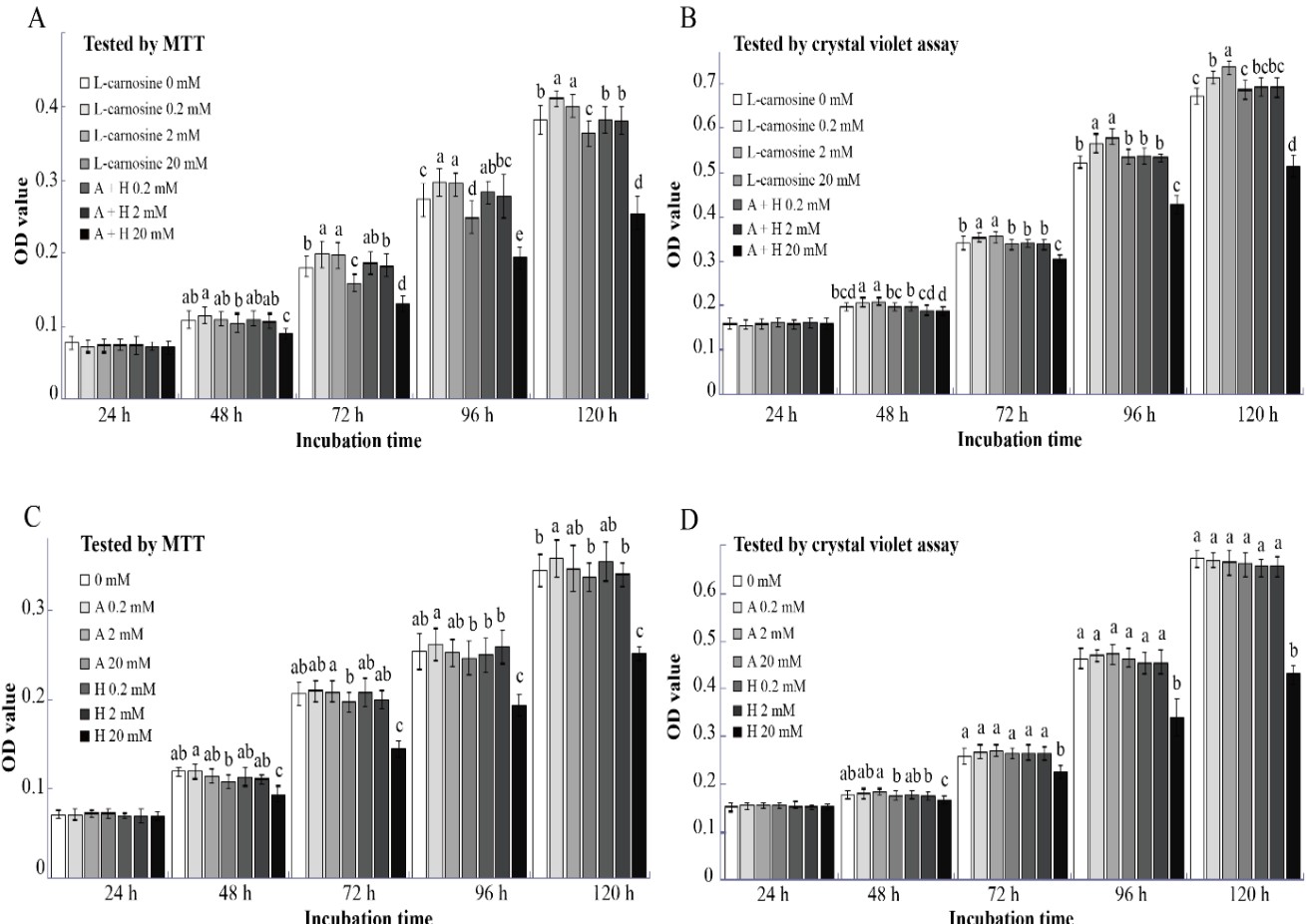

**Figure 3.** Effect of L-carnosine constituents on the proliferation of SCs in vitro (*n* = 12). The effect of L-carnosine and the combination of β-alanine and L-histidine on SC proliferation (**A**,**B**). The effect of β-alanine and L-histidine on SC proliferation (**C**,**D**). MTT, 3-(4,5-dimethylthiazol-2-yl)-2,5-diphenyltetrazolium bromide; A, β-alanine; H, L-histidine; L-carnosine (0, 0.2, 2, and 20 mM), growth medium supplemented with different concentrations of L-carnosine (0, 0.2, 2, and 20 mM); A+H (0.2, 2 and 20 mM), growth medium supplemented with different concentrations of β-alanine and L-histidine (0.2, 2, and 20 mM); 0 mM, growth medium without β-alanine and L-histidine; A (0.2, 2, and 20 mM), growth medium supplemented with different concentrations of β-alanine (0.2, 2, and 20 mM); H (0.2, 2, and 20 mM), growth medium supplemented with different concentrations of L-histidine (0.2, 2, and 20 mM). Data are expressed as mean values ± SD. Means with different superscript letters (a–e) indicate significant differences at *p* < 0.05.

### 3.4. Effect of L-Carnosine on Cell Size and Cell Cycle of Satellite Cells

As cell division is regulated by cell cycle [29], the effect of L-carnosine on the proliferation of piglets' SCs can be attributed to its cell-cycle activity. Cell size and cell-cycle distribution was measured by means of FCM. Cell size was not affected (*p* > 0.05) due to L-carnosine supplementation when compared to the control treatment (Figure 4A). The distribution of DNA content of cells consisted of two peaks corresponding to the cells in resting (G0)/first gap (G1) and second gap (G2)/mitotic (M) phase, respectively. Cells in the synthesis (S) phase were located between the two peaks (Figure 4B). At 72 h after seeding, SCs incubated with 2 mM of L-carnosine had a lower (*p* < 0.05) proportion of cells in the G0/G1 phase, and a higher (*p* < 0.05) proportion of cells in the S phase in comparison to the control treatment (Figure 4C). Compared to the control treatment, SCs incubated with 0.2 and 20 mM of L-carnosine had a higher (*p* < 0.05) percentage of cells in the S phase

(Figure 4C). Cell population in the G2/M phase was not affected ($p > 0.05$) by supplemental L-carnosine (Figure 4C).

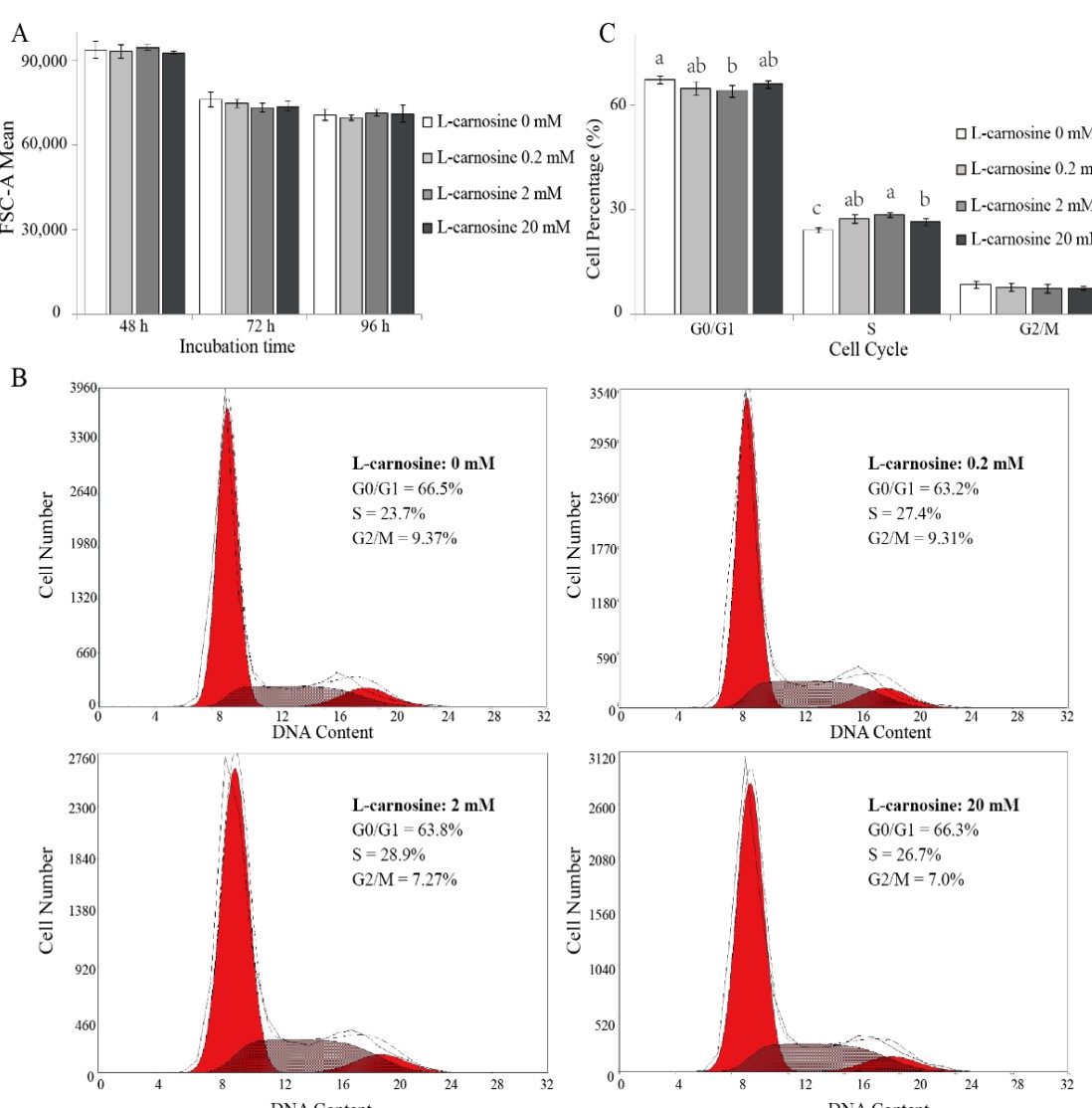

**Figure 4.** Effect of L-carnosine on cell size and cell cycle in piglets' SCs in vitro ($n = 5$). The effect of L- carnosine on SC size (**A**) and cell-cycle distribution at 72 h (**B**,**C**). L-carnosine (0, 0.2, 2, and 20 mM), growth medium supplemented with different concentrations of L-carnosine (0, 0.2, 2, and 20 mM). FSC-A, forward scatter area; G0, resting phase; G1, first gap; S, synthesis phase; G2, second gap; M, mitotic phase. The data are expressed as mean values ± SD. Means with different superscript letters (a–c) indicate significant differences at $p < 0.05$.

### 3.5. L-carnosine Activated IGF-1/PI3K/S6K Signaling Pathway in Cultured Cells

Owing to the important role of the mTOR pathway for cell proliferation and growth and protein synthesis, the expression of key regulators of the mTOR pathway was determined. At 72 h after seeding, the expression of IGF-1 (Figure 5A), phosphorylation ratios of Akt (Ser473) (Figure 5B), mTOR (ser2448) (Figure 5C), and S6K (Thr389) (Figure 5D) was higher ($p < 0.05$) in GM with 2 mM of L-carnosine in comparison to the control treatment devoid of L-carnosine. There were no differences for p-4EBP1 (Thr70) (Figure 5E) or eIF4E (Figure 5F) ($p > 0.05$) between the control and L-carnosine treatments. In addition, values for Akt, mTOR, S6K, and 4EBP1 between the control and L-carnosine treatments did not differ ($p > 0.05$) (data not shown).

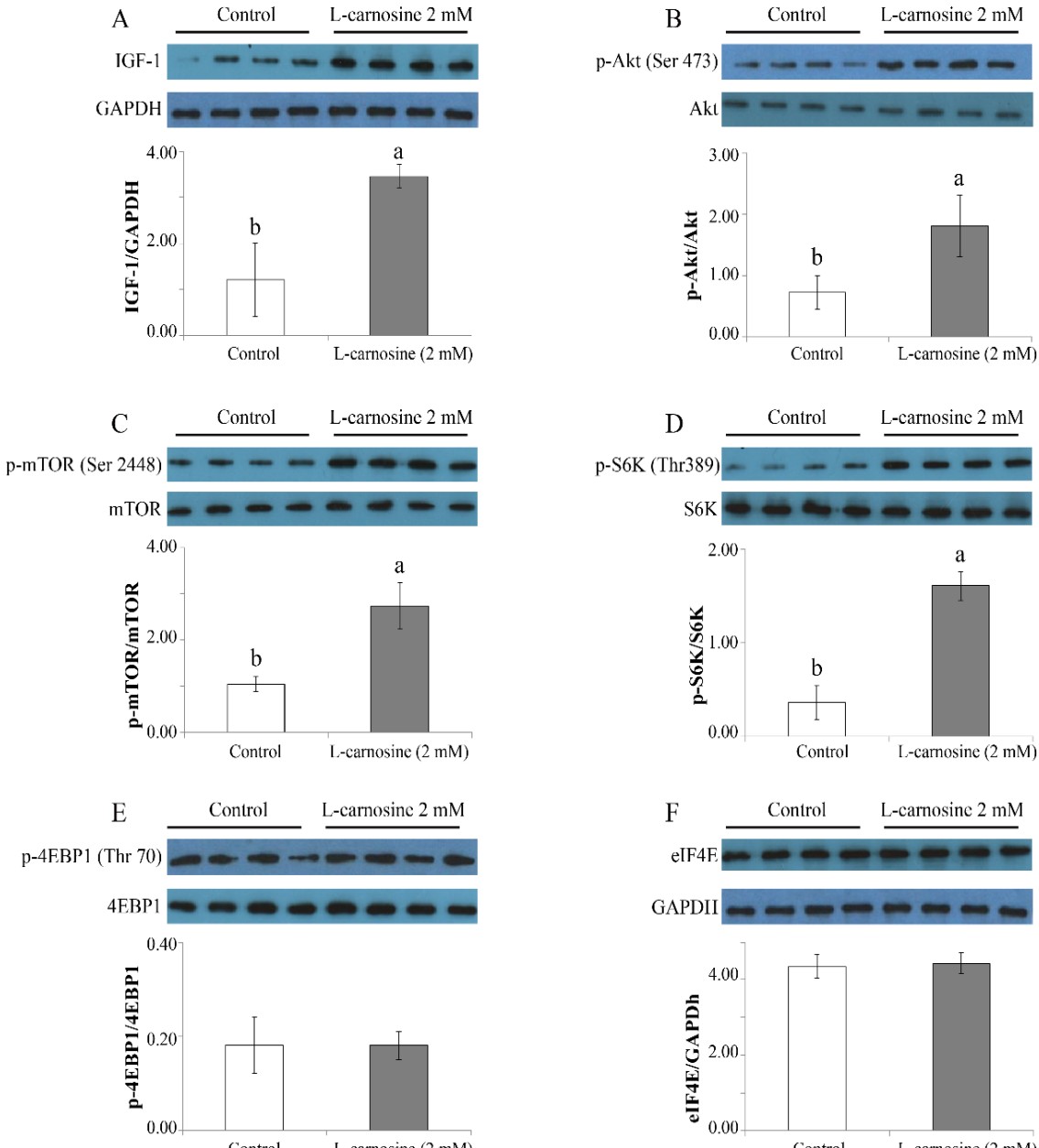

**Figure 5.** Effect of carnosine on expression of mTOR pathway in satellite cell cultures (*n* = 4). Values of IGF-1 (**A**) and eIF4E (**F**) are presented in relation to the GAPDH concentration, while the values of Akt (**B**), mTOR (**C**), S6K (**D**), and 4EBP1 (**E**) are presented as phosphorylation concentrations in relation to the total concentration. Control, growth medium without L-carnosine; L-carnosine 2 mM, growth medium received 2 mM L-carnosine. p-, phosphorylation; Ser, serine; Thr, threonine; IGF-1, insulin-like growth factor-1; Akt, protein kinase B; mTOR, mechanistic target of rapamycin; S6K, ribosomal protein S6 kinase; 4EBP1, 4E binding protein 1; eIF4E, eukaryotic initiation factor-4. The data were expressed as mean ± SD. Means with different letters (a, b) indicate significant difference at $p < 0.05$.

## 4. Discussion

In this study, the possible effect of a diet supplemented with L-carnosine on growth performance of pig was evaluated and the underlying mechanisms were explored. In line with the working hypothesis, ADG of pigs was substantially improved upon oral intake of L-carnosine. In addition, for the L-carnosine treatment, there was a tendency for enhanced

growth of the pigs' loin eye muscle area. In agreement with previous reports, dietary L-carnosine supplementation increased ADG and ADFI by 7.60 and 9.00%, respectively, compared to pigs fed diets devoid of L-carnosine [21]. Similarly, dietary supplementation of L-carnosine at a level of 0.5% in a diet for broilers increased birds' breast and thigh muscle weight and dressing percentage, and there was a trend toward higher ADG [22]. Whereas, Ma et al. (2010) observed no effect of dietary L-carnosine supplemented at a level of 0.01% to the diet of finishing pigs on pigs' growth performance. It cannot be ruled out that the considerably lower supplementation level of L-carnosine in this study compared to the aforementioned studies in birds and pigs is responsible for the observed discrepancy.

It has been well established that the mTOR pathway, which can be activated by nutrients [30] and growth factors [31], represents an essential positive regulatory process for skeletal muscle growth [15,32]. However, the underlying mechanism of nutrients affecting the mTOR pathway and thereby promoting muscle growth are not fully understood. The relatively high expression of p-Akt and p-mTOR upon activation by nutrients may result in greater muscle mass in lambs [33]. This observation is in agreement with our finding that the related protein in the mTOR pathway, p-Akt and p-mTOR, was activated in the *longissimus dorsi* muscle due to L-carnosine supplementation. Activation of the Akt-mTOR pathway may upregulate the S6K, thus promoting protein synthesis [15] and growth of skeletal muscle mass [34].

The present study points toward a trend such that supplemental carnosine may promote the magnitude of *longissimus dorsi* muscle growth and the proliferation of SCs, in addition to enhanced cell activity and cell cycle, together with the activation of the Akt/mTOR pathway in both muscle and cells. Furthermore, under in vivo conditions, L-carnosine can activate the mTOR signaling pathway, which was confirmed by using skeletal muscle SCs. The activation and proliferation of SCs are required for muscle regeneration and growth [35]. Proliferation has been characterized as a process by which cell numbers are increased through mitosis [36]. In the present study, treatment with 0.2 or 2 mM L-carnosine increased the proliferation rates of SCs of pigs in vitro. This effect may be attributed to the function of carnosine as an activator of cell activity [37]. It can be derived from the results of the in vitro study that lower concentrations of L-carnosine are in favor of promoting SC proliferation, whereas higher concentrations exerted adverse effects on the proliferation rates of SCs. This finding is consistent with the results of other studies where the high concentration of L-carnosine (50 mM) inhibited cell proliferation and cell-cycle arrest in G0/G1 in human gastric carcinoma cells [38], mesangial cells [39], and myoblast cells [40]. In the present study, the ratio of G0/G1 cells was decreased in GM supplemented with L-carnosine at 2 mM, while an increased ratio of S phase cells at concentrations of 0.2 and 2 mM L-carnosine was obtained. However, cell size was not affected by supplemental L-carnosine. These results suggest that enhanced proliferation of SCs by L-carnosine may reactivate the G0/G1 phase of the cell cycle.

Besides regulation of the mTOR pathway by means of nutrients [30], growth factors can also be involved [31]. For example, the IGF-1 acts as an autocrine/paracrine mediator in muscle processes, and plays a key role in proliferation and differentiation of SCs and muscle growth and development [41]. This is in line with the present results of the in vitro study, where an increase in IGF-1 was observed upon supplementation of 2 mM L-carnosine. Upon binding to the IGF receptor, IGF-1 activates the IGF receptor, thereby regulating the mTOR pathway [42]. Measurements of the mTOR pathway may provide further insight into the regulatory mechanisms of L-carnosine concerning differentiation of SCs and skeletal muscle growth. Akt/mTOR has been identified as important regulator of skeletal muscle hypertrophy in association with increased protein synthesis [34]. Ginsenoside Rb1 and Rb2, the most abundant ginsenosides in *Panax ginseng*, can enhance myoblast proliferation and muscle growth via activating the Akt/mTOR signal pathway in vitro [43]. In addition, Rab5, a small guanosine triphosphate enzyme, has been shown to increase activation of the Akt/mTOR signal pathway during skeletal muscle growth in vivo [44]. In agreement with the present findings, 2 mM and 0.1% of L-carnosine considerably increased levels of IGF-1,

p-Akt, p-mTOR, and p-S6K in vitro and in vivo, respectively. These results suggest that L-carnosine improved the SC proliferation rate via activating the IGF-1/Akt/mTOR/S6K signal pathway. The vital role of mTOR in SC function as shown in the present study has been confirmed in previous studies as well [12,32,41].

## 5. Conclusions

Dietary supplementation with 0.1% L-carnosine improved piglets' growth performance, and GM with low concentrations of L-carnosine (0.2 and 2 mM) increased proliferation and percentage of SCs in the S phase, decreased the percentage of SCs in the G0/G1 phase, and activated the mTOR signal pathway both in *longissimus dorsi* of pigs and cultured cells. The SCs are closely related to muscle growth, and it can be assumed that the improved growth performance upon oral intake of L-carnosine may be due to the enrichment of L-carnosine elevated in muscles, thereby promoting SC proliferation through an activated IGF-1/Akt/mTOR/S6K signaling pathway, resulting in increased muscle growth of pigs.

**Author Contributions:** Y.L.: conceptualization, conducted the cell experiments, and writing—original draft. W.S.: methodology and writing—review and editing. T.L.: data curation. R.M.: writing—review and editing. P.C.: conducted the animal experiments. Y.B. and W.H.: assisted with the experiments. L.Z., J.Z. and C.J.: supervision. Q.M.: conceptualization, funding acquisition, project administration, and writing—review and editing. All authors have read and agreed to the published version of the manuscript.

**Funding:** This study was funded by National Natural Science Foundation of China (Grant No. 31572447), China Program for New Century Excellent Talents in University (NCET-13-0558), and a Sino-German cooperative fund between China Agricultural University and Hohenheim University (CAU/UOH 201704810410317).

**Institutional Review Board Statement:** All of the animal experimental procedures were handled in accordance with the guidelines of Beijing on the Review of Welfare and Ethics of Laboratory Animals, and approved by China Agricultural University (Beijing, China) Animal Care and Use Committee (Approval No. AW13301202-1-4).

**Informed Consent Statement:** Not applicable.

**Data Availability Statement:** Data are available from the first author.

**Conflicts of Interest:** The authors declare no conflict of interest.

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
