# Peer review of "Improved Satellite Cell Proliferation Induced by L-Carnosine Benefits Muscle Growth of Pigs in Part through Activation of the Akt/mTOR/S6K Signaling Pathway"

_agriculture, doi:10.3390/agriculture12070988_

Round 1
Reviewer 1 Report
General comments:
The paper is interesting. However, some points in the material and methods require more details to clarify the experimental design. The dietary protein level is too high for 50-kg pigs. Please clarify what requirement standard has been used. Because this diet maintains more than Landrace pigs amino acids requirement. Particularly that carnosine is naturally produced by the body in the liver from beta-alanine and histidine and this extra amino acid may facilitate carnosine production in the liver. So, it's confusing how dietary supplementation of carnosine can be effective. Please clarify.
It is not clear how many one-day-old pigs were used for obtaining skeletal muscle SCs. Please clarify how the corresponding pigs reached 75 kg in 21 days (line 91-92; 1.190 kg/d), but their ADG was around 0.8 kg/d (Table 2).
Tables and figures have to be self-contained. Please modify them.
Table 2, please clarify what does “ALA, β-alanine, HIS, L-histidine; ALA+HIS, β-alanine, and L-histidine” mean.
Figure 2, please define “MTT”
Figure 3 please define all the treatments in the figure legend.
Figure 4 please define FSC-A
Figure 5 please define all the abbreviations
In the conclusion, the authors have to clearly explain what carnosine level was the best rather than writing “low concentration”.
Reviewer 2 Report
The title as well as keywords accurately reflects the major findings of the work.
The abstract adequately summarize methodology, results, and significance of the study. However, Authors should indicate statistical analysis applied on the data.
The introduction section falls within the topic of the study, however, Authors should enhance this section adding more information concerning the breeding strategy studies focused on the improvement of animal reproduction and production and on the diet supplementation in veterinary field emphasizing the significant increase of interest showed by scientific community on diet improvement to enhance animal health status and welfare. On this regard, I suggest to remove the sentences (Lines 59-64) “L-carnosine, a naturally occurring dipeptide, which is composed of β-alanine and L- histidine, is widely distributed in tissues, and exists at relatively high concentrations in skeletal muscles and brain [16]. L-histidine is an essential amino acid for non-ruminants, while β-alanine is a non-essential amino acid and is the rate-limiting precursor for synthesis of L-carnosine [17]. L-carnosine can improve calcium binding ability [18, 19], and also plays an important buffering role in skeletal muscles [20].” from introduction section. Authors could move these sentences in discussion section. in place of these sentences, I suggest adding “Overall, feed additives, nutrients that when added to the feed can trigger the desired response of the animal's body on production parameters, have been widely used in animal nutrition for quite a long time. In recent years, nutritional strategies have emerged and it has been well demonstrated that they can increase the level of production and improve the health of animals and products obtained from them (Abbate J.M. et al., Animals 2020, 10, 2303) as well as to enhance productivity in livestock (Avondo M. et al., Journal of Dairy Research, 2009, 76: 202-209; Vazzana I. et al., Comparative Clinical Pathology, 2014, 23: 1587-1591; Armato L. et al., Acta Agriculturae Scandinavica, Section A, 2016, 66: 119-124).”
I suggest to change “In this study,… pathway in cultured cells was studied” (Lines 74-78) with “In view of such considerations, the aim of the current study was to evaluate potential effects of L-carnosine on growth performance and on the expression of pivotal regulatory factors of the mTOR pathway in longissimus dorsi muscle of pigs. In addition, the possible effect of a diet supplemented with L-carnosine on SCs proliferation, cell cycle of SCs, and mTOR signaling pathway in cultured primary pig skeletal muscle SCs culture.”
The section of Materials and Methods is clear for the reader however, some clarifications are needed and some missing information should be added.
Did Authors evaluate the health status of enrolled animals?
Authors should indicate the age of enrolled animals.
Regarding sampling, Authors wrote “For all data, conformity to normal distribution was determined using the Shapiro- Wilk test.” Please specify whether data passed normality test indicating the P value.
Results section as well as Discussion section is clear and well written. The findings obtained in the study were well discussed and justified with appropriate references. I suggest to add an introducing sentence (or brief paragraph) at the beginning of the discussion section in order to briefly introduce to topic of the study.
The conclusion section is well written, indeed, Authors well summarize the results and the significance of the study. Please delete “In conclusion,”
The tables and figures are generally good and well represent the results of the study.
Authors should check and standardize the references in the list according to journal guidelines.
Round 2
Reviewer 1 Report
The manuscript has been corrected well and I have no more comments.